# Refinement of Landslide Susceptibility Map Using Persistent Scatterer Interferometry in Areas of Intense Mining Activities in the Karst Region of Southwest China

**Chaoyong Shen [1,2]**, **Zhongke Feng [1,\*]**, **Chou Xie [3,4]**, **Haoran Fang [4]**, **Binbin Zhao [5]**, **Wenhao Ou [5]**, **Yu Zhu [6]**, **Kai Wang [7]**, **Hongwei Li [7]**, **Honglin Bai [7]**, **Abdul Mannan [1]** and **Panpan Chen [1]**

[1] Precision Forestry Key Laboratory of Beijing, Beijing Forestry University, Beijing 100083, China; shency@gzchsy.cn (C.S.); mannan@bjfu.edu.cn (A.M.); chenpanpan@bjfu.edu.cn (P.C.)
[2] The Third Surveying and Mapping Institute of Guizhou Province, Guiyang 550004, China
[3] Key Laboratory of Microwave Objective Characteristics and Remote Sensing in Zhejiang Province, Zhongke Satellite Application Deqing Academy, Deqing 313200, China; xiechou@radi.ac.cn
[4] Institute of Remote Sensing and Digital Earth, Chinese Academy of Sciences, Beijing 100101, China; 09231258@bjtu.edu.cn
[5] China Electric Power Research Institute Co., Ltd., Beijing 100055, China; zhaobinbin@epri.sgcc.com.cn (B.Z.); owh@epri.sgcc.com.cn (W.O.)
[6] School of Earth Sciences and Resources China University of Geosciences, Beijing 100083, China; 2101170099@cugb.edu.cn
[7] School of Civil Engineerin, Beijing jiaotong University, Beijing 100044, China; 17121175@bjtu.edu.cn (K.W.); 17121173@bjtu.edu.cn (H.L.); 16121173@bjtu.edu.cn (H.B.)
\* Correspondence: zhongkefeng@bjfu.edu.cn

**Abstract:** Incremental vertical ground movements due to coal mining can increase landslide susceptibility greatly in a short time and have thus triggered a large number of geological disasters, especially in the Karst Region, where a lot of steep slopes are on fractured rocks. Therefore, the landslide susceptibility maps (LSM) in Karst Region should be updated regularly. This paper presents an efficient methodology to update and refine LSM by using Persistent Scatterer Interferometry (PSI) data directly. First, an original LSM was produced by using a support vector machine (SVM) algorithm, and the distribution of coal mining was considered a crucial factor to generate the LSM. Then, the Permanent Scatterer Interferometric Synthetic Aperture Radar (PSInSAR) technique was implemented to retrieve displacement time-series. Finally, the landslide displacement map, produced by the PSInSAR analysis, was projected to the direction of the steepest slope and resampled to the same cell in the LSM, to update the original LSM. This methodology is illustrated with the case study of Bijie in the Karst Region of Southwest China, wherein the ascending RADARSAT-2 and descending Sentinel-1 datasets are processed for the period of 2017–2019. The results show that the susceptibility degree increased in 56.41 km$^2$ of the study area, and 80 percent of the increased susceptibility degree was caused by coal mining. The comparison between original and refined LSM in two specific areas revealed that the proposed method can produce more-reliable landslide susceptibility maps in areas of intense mining activities in the Karst Region.

**Keywords:** landslide susceptibility; Persistent Scatterer Interferometry; intense mining activities; Karst Region

---

## 1. Introduction

Landslides constitute a serious source of danger, causing environmental damage and substantial human and financial losses. Therefore, landslide monitoring, risk assessment, and prediction are urgent subjects for the international landslide disaster research community. Landslide susceptibility mapping can provide information about the spatial distribution of the probability of regional slope instability [1,2], which is the first and the most important step in landslide hazard assessment, in order to take effective measures for landslide mitigation. In recent years, the rapid development of Geographical Information Systems (GIS) technology and mathematical statistics tools has promoted the wide application of methods for the quantitative evaluation of landslide susceptibility.

The weights of evidence [3,4], artificial neural network [5,6], random forest [7–9], and support vector machine [10,11] models have been used for the elaboration of landslide susceptibility or hazard maps [12]. The major impact factors that can affect landslide susceptibility include terrain, geology, geomorphology, and land cover [13]. These factors seldom undergo great changes in the span of one or two years; meanwhile, landslide risk increases significantly in areas of intense human activity. To increase the reliability of the LSM, some dynamic information, such as regional displacement information produced by Interferometric Synthetic Aperture Radar (InSAR) data, should be included for the dynamic updating of LSMs.

Persistent Scatterer Interferometry (PSI) techniques are widely employed in geosciences to detect and monitor landslides, with high accuracy over large areas in three different stages of landslide studies [14]: (i) the identification of individual landslides and the delineation of generalized unstable areas [15,16]; (ii) the detection of potential landslides over a large area based on surface deformation maps [17,18]; and (iii) the mapping of automatic slow-moving landslides based on the application of a spatial statistical approach [19,20]. PSI-based ground displacement can be utilized to assess landslide susceptibility at the regional scale, using the following steps: (i) the comparison of the persistent scatterer (PS) distribution in each class of the landslide susceptibility map based on landslide inventories [21]; (ii) the exploratory assessment of landslide susceptibility based on the PSs [22]; and (iii) the updating of landslide hazard and risk maps based on Persistent Scatterer Interferometry [23].

Mining in karst terrains usually imposes additional subsidence effects on the sloping ground surfaces and increases the tendency of deformations angling toward valleys and slopes [24,25], which may finally trigger landslides. Previous works seldom use PSI and related techniques to refine landslide susceptibility maps in areas of underground coal mining. In this study, we present an efficient methodology to update and refine LSM by using PSI data in areas of intense mining activities in the Karst Region. The paper is organized as follows. Section 2 describes landslide susceptibility in Bijie, Guizhou Province, China, and recounts the collection of Synthetic Aperture Radar (SAR) data and geological data, respectively; Section 3 explains the method adopted to generate and update the landslide susceptibility map with the assistance of the PSI technique; and Section 4 is dedicated to the presentation of the obtained results. A discussion and further developments are included in Sections 5 and 6.

## 2. Study Area and SAR Data

### 2.1. Study Area

The study area is Bijie, which expands across 26,853 km$^2$ in the northwestern part of Guizhou Province (Figure 1). This region is one of the karst geomorphology distributing areas in Southwest China. Bijie has a largely distributed karst area, accounting for 72.8% of the total area [26]. Tectonically, Bijie has a complicated geological setting due to extensive faulting [27]. The key tectonic changes controlling the tectonostratigraphic framework of Bijie and the adjacent areas are the Duyun, Guangxi, Central Indosinian, Late Yanshanian, and Himalayan movements. Three types of caprocks can develop in Bijie: salt rock, mudstone and shale, and condensed sandstone. The regional caprocks with good reservation are mudstone and shale of the Lower Cambrian and Mid-Lower Silurian, as well as

limestone and marl of the Lower Cambrian. The topography of Bijie varies greatly, with a maximum elevation of 4855 m and a minimum elevation of 144 m. Western Bijie is a tableland with a higher average elevation (about 3600 m) than eastern Bijie. Severe rain events always occur in the study area from May to September, and the average monthly rainfall of these months is above 100 mm. The maximum monthly rainfall reaches up to 300 mm. (Figure 2).

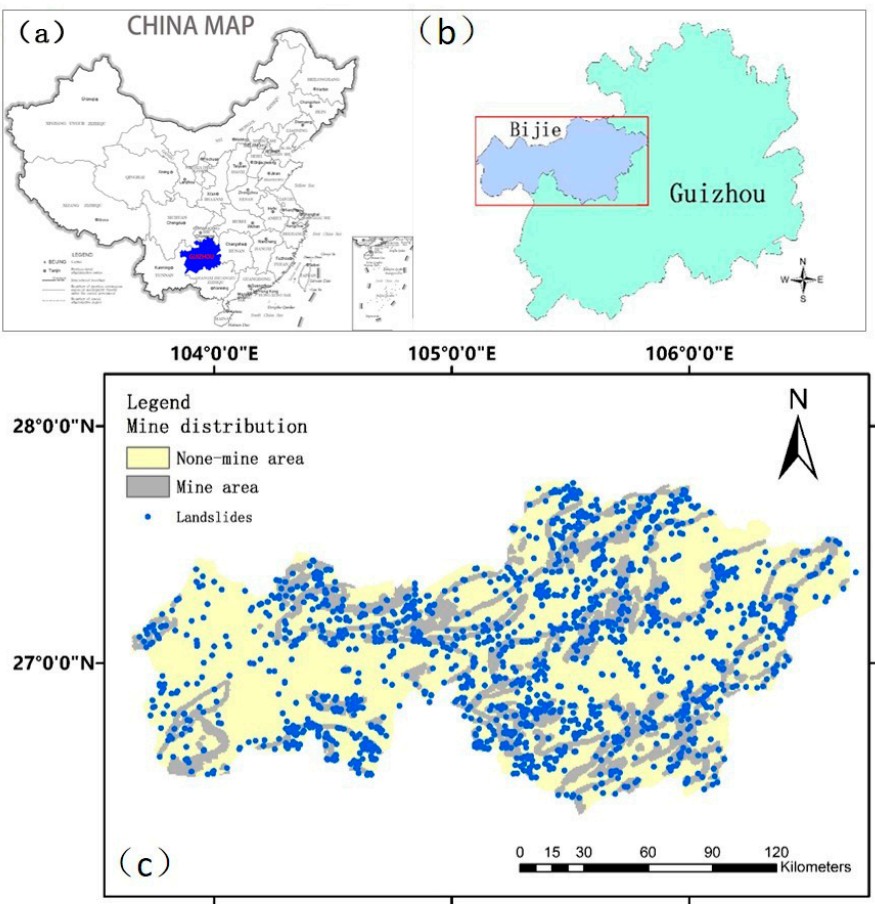

**Figure 1.** (**a**,**b**) Location of the research region in China; (**c**) distribution map of the coal mine area and landslides in Bijie.

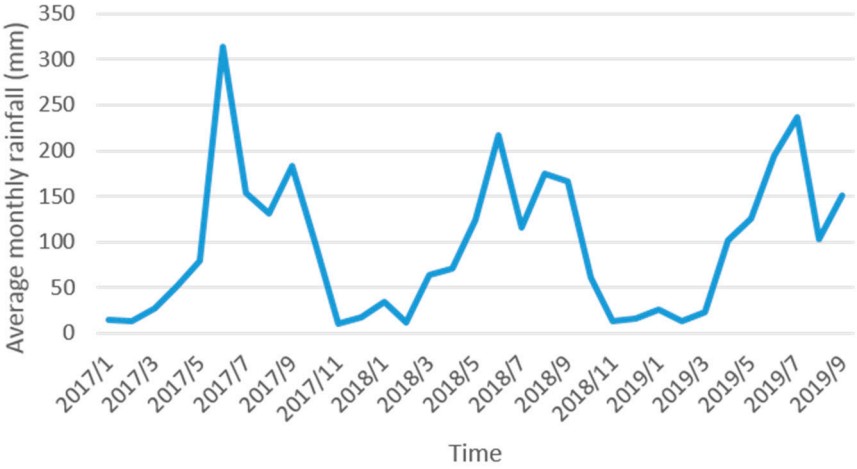

**Figure 2.** Average monthly rainfall of Bijie from 2017 to 2019.

Western Guizhou Province contains high-quality anthracite coal, an important resource for China, and Bijie is a major coal-producing region in Guizhou Province [28,29]. Its coal resources are distributed over all the eight counties in the whole region of Bijie. At present, Bijie has 4.492 billion tons of proven coal reserve, accounting for 45.9% of the total reserves in Guizhou Province [30]. The coal-bearing area reaches 12,000 km², accounting for 45% of the total land area of Bijie. The distribution of the coal-bearing area is not homogeneous in Bijie; 78% of the coal bearing area is distributed in Eastern Bijie. Most of the coal seams in the area are produced by near-horizontal (dip angle below 8°) and gently inclined (dip angle between 8° and 25°) coal seams.

Coal-mining operations lead to incremental vertical ground movements, which may cause the lateral dilation of the subsiding rock strata (Figure 1c). Moreover, exceptionally high rainfall events combined with the presence of steep slopes on fractured rocks have triggered a large number of geological disasters. In Bijie, landslides are a recurrent phenomenon, causing loss of human life, significant economic damage to buildings and infrastructures, and loss of productive soils and pasture lands. There are 457 potential landslide sites distributed in Bijie, and 237 thousand people live under the threat of geological hazards. On 28 August 2018, a large landslide occurred in Nayong County, Bijie, and the volume of the rock fall was estimated to be ~600,000 cubic meters. Seventeen people died, nine people were injured, and 170 houses were completely destroyed in that disaster.

## 2.2. SAR Data

Figure 3 depicts the extension of RADARSAT-2 and Sentinel-1 SAR scenes. We used over 130 SAR images of four ascending tracks acquired by the RADARSAT-2 satellite and 41 SAR images from one descending track acquired by the Sentinel-1 satellite between 2017 and 2019, covering an area of 26,853 km² in Bijie. This coverage allowed us to investigate displacement over the entire study area, and the combination of ascending and descending datasets provides a chance to solve topographic effect problems and make the majority of slopes visible from the opposite pair. RADARSAT-2 has a repeated orbital period of 24 days, and Sentinel-1 has a repeated orbital period of 12 days. For our 2.5 year period, we used an average of 23 RADARSAT-2 and 41 Sentinel-1 acquisitions per location.

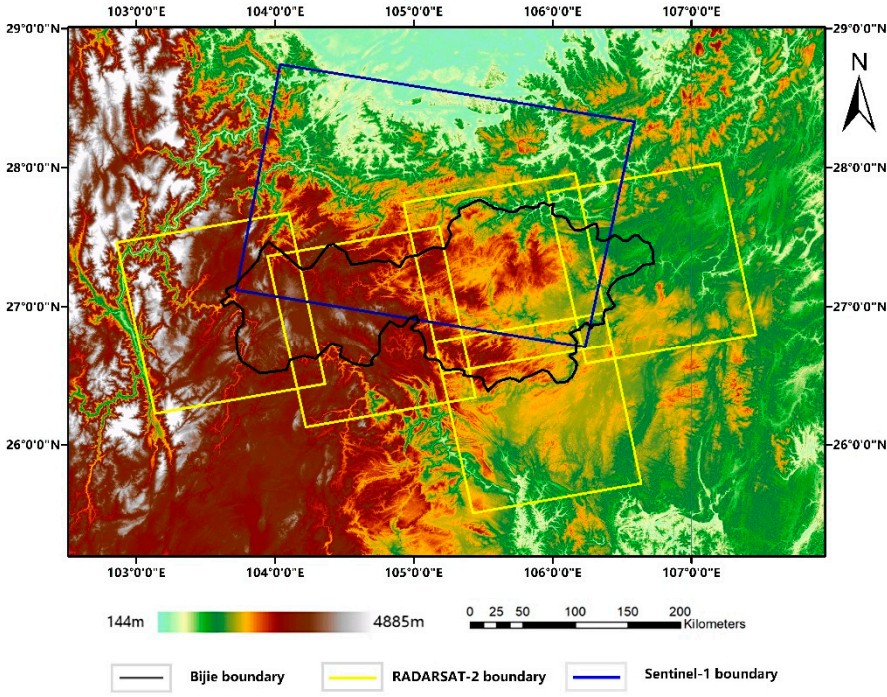

**Figure 3.** RADARSAT-2 and Sentinel-1 coverage and topography of Bijie.

In Table 1, the main acquisition characteristics of the SAR dataset used in this work are shown. The obtained RADARSAT-2 SAR data are from the Extra Fine mode, offering a resolution of 5 m on ground and a swath width of 120 km. The obtained Sentinel-1 SAR data are from the Interferometric Wide (IW) swath mode, offering a resolution of 20 m on ground and a swath width of 240 km. The incidence angles of RADARSAT-2 and Sentinel-1 are 35.2° and 33.9°, respectively.

**Table 1.** Main acquisition parameters of RADARSAT-2 and Sentinel-1 SAR datasets in Bijie.

| Sensor | RADARSAT-2 | Sentinel-1 |
|---|---|---|
| Band | C | C |
| Acquisition orbit | Ascending | Descending |
| Incidence angle (°) | 35.2 | 33.9 |
| Swath width (km) | 125 | 240 |
| Repeat cycle (days) | 24 | 12 |
| Number of images | 130 | 41 |
| Ground resolution (m) | 5 | 20 |
| Temporal range | 9 April 2017–28 July 2019 | 4 January 2018–24 July 2019 |

Due to the layover, foreshortening, and shadowing effects that were caused by the side-looking geometry of SAR, some of the areas of interest do not have adequate geometrical visibility for displacement measurement by PSI. However, this effect can be mitigated by ascending–descending orbit combination. Concretely, we used the ascending RADARSAT-2 and descending Sentinel-1 datasets to improve the geometrical visibility of the whole study area.

## 3. Methodologies

### 3.1. Landslide Susceptibility Map

A support vector machine (SVM) algorithm in the sklearn package of Python was implemented to produce the landslide susceptibility map (LSM) of Bijie. As a machine-learning algorithm, SVM has been widely employed in different classification and regression problems because of its effectivity in working with linearly non-separable and highly dimensional datasets [31]. SVM has been proven to be a valid algorithm for the classification of remote-sensing data—for example, for target recognition and land classification. The performance of the SVM model is greatly affected by the adopted kernel functions, such as linear, polynomial, sigmoid, and radial basis function (RBF, often called Gaussian) kernels. In this work, RBF was chosen as the kernel function, and the SVM model was built by using the training data and then applied to calculate the landslide susceptibility indices for the entire study area.

The LSM of Bijie was constructed, taking into account 12 conditioning factors. These factors were considered in the analysis of landslide susceptibility based on a literature review and data availability [32,33]. They include curvature, altitude, slope angle, slope aspect, distance to stream [34], plan curvature, profile curvature, mining disturbances, distance to road, lithology, rainfall, and land use. The 30 m digital elevation model (NASA Shuttle Radar Topography Mission (SRTM) Global 1 arc second product) was used to calculate the slope angle, slope aspect, plan curvature, and profile curvature [35]. Topography is an important feature in the creation of landslides. The precipitation map is based on meteorological data. The monthly mean precipitation is as shown in Figure 2. The lithology is based on the lithology map in Bijie. Lithology is an important factor of landslide, and different lithology plays an important role in slope deformation. The road and river data were acquired from Bijie's geographic database. River erosion and road construction will make the mountain produce the empty face of slope. Mining disturbances, which were digitized from the coal-resources planning maps, were considered to be a crucial factor when generating the LSM in Bijie. Landslide inventory datasets are also a key factor in the generation of the LSM and the prediction of future accident risks.

Landslide susceptibility mapping is an effective method that reflects the risk of landslides in the study area. In this study, 12 factors were extracted to generate the LSM based on the information about

the region of Bijie. Subsequently, the LSM was classified into four classes: (i) low to null susceptibility, (ii) moderate susceptibility, (iii) high susceptibility, and (iv) very high susceptibility. The landslide problem is typically a two-class model.

For two-class SVM models, it is necessary to generate samples, as landslide modeling is considered a form of binary-pattern recognition. According to the results of the geological survey, there were 1677 landslide points in Bijie. These landslide points were randomly sampled and divided into two groups. One group was used for training, and the other was used for validation. A further 1000 non-landslide points were also selected in Bijie, and also divided into two groups (training and validation sets). Landslide and non-landslide points were combined to form the merged training and validation sets.

### 3.2. PSInSAR

The Persistent Scatter Interferometry (PSI) and Small Baseline Subsets (SBAS) techniques, which monitor displacement time-series by using successive InSAR observations, have led to significant progress in InSAR technology [36,37]. In recent years, a new approach called Distributed Scatter Interferometry (DSI) has been proposed to retrieve displacement time-series from natural targets [38,39]. Distributed scatterers (DSs) correspond to homogeneous areas spread across a group of pixels in an SAR image (e.g., rangeland, pasture, shrubs, and bare soils). The DSI technique considerably increases the point target density compared to that of the traditional PSI technique, especially in sparsely vegetated landscapes [40–42]. The DSI technique certainly represents one of the most recognized methods to measure, with unprecedented accuracy, landslide displacements at a regional scale.

Our basic DInSAR processing, based on the InSAR Scientific Computing Environment (ISCE) software, includes geometric calibration of the master image, co-registration of the single-look complex image and interferogram generation. Image calibration, which involves the estimation of the near slant range and the time of acquisition of the first image line, is based on an external Digital Elevation Model (DEM) of the observed area. The calibration is followed by image co-registration and generation of a redundant network of M interferograms from N collected images (M ≫ N) according to interferogram normal baselines and temporal baselines. In this work, we generated RADARSAT-2 interferograms with a maximum temporal baseline of 120 days and Sentinel-1 interferograms with a maximum temporal baseline of 60 days. After the removal of the topographic-related phase with use of the external DEM, the more suitable PS candidates of an interferogram stack were selected, using the amplitude dispersion index and spectral characterization. In addition, a two-sample Anderson–Darling (AD) test was used to identify statistically homogenous pixels (SHP). Pixels with >20 SHP were assumed to be DS candidates. A coherence-weighted phase-linking method was adopted for phase optimization before DS detection. The selected pixels are connected pair-wise by edges, computing the phase difference for each edge and for each of the M interferograms. For each edge of the network a phase difference was computed by differentiating the phases of the two corresponding pixels. Then, the differential terrain deformation and the differential topographic error was estimated at each edge. Finally, these differential values were integrated into the entire set of selected pixels. The most relevant stages of the procedure are briefly described below.

The atmospheric effects, nonlinear deformation, and possible orbital track inaccuracies either occur on a very large scale or are small in amplitude only, which can be strongly mitigated by the spatial differentiation operation. Under this assumption, for each phase difference, we can use the following equations:

$$\Delta\varepsilon^k = \Delta\varnothing^k_{obs} - \Delta\varnothing^k_m \tag{1}$$

$$\Delta\varnothing^k_m(\Delta h, \Delta v) = \frac{4\pi}{\lambda}\frac{B^k_\perp}{R^k sin\theta}\Delta h + \frac{4\pi}{\lambda}\Delta T^k \Delta v \tag{2}$$

where $\Delta\varepsilon^k$ is the differential phase residual associated with a given edge, e, and $\Delta\varnothing^k_m$ is the modeled differential phase. $\Delta h$ and $\Delta v$ are the two unknown difference topographic errors and deform velocity associated with the edge, e, respectively. In practice, these edges must be as short as possible in order

to minimize the approximation effects in the phase difference. Several methods to connect an irregular set of points can be used, e.g., the Delaunay triangulation. In this study, redundant connections (edges) between pixels are used because they provide more robust results.

At first, we constructed the skeleton of network by Delaunay triangulation and maximized the temporal coherence function on each edge:

$$\gamma(e) = \left| \frac{1}{M} \sum\nolimits_{k=1}^{M} exp\left(j\Delta\varepsilon^k\right) \right| \tag{3}$$

To find the values $\Delta\hat{h}$ and $\Delta\hat{v}$ that give the maximum $max(\hat{\gamma})$, the method of the periodogram is used. If the maximum $max(\hat{\gamma})$ is larger than a given threshold, i.e., $T_\gamma = 0.6$, we preserve the edge; if otherwise, we reject it. In order to use the temporal coherence resulting from this estimation on such edges to eliminate any useless isolated points. A successive Delaunay triangulation is implemented once more to refresh the spatial edges distribution over longer distances and a further $\Delta\hat{h}$ and $\Delta\hat{v}$ estimation is again carried out, until there are isolated points. This procedure assures that any point on the skeleton of network has high-quality edge connection. Then we intensify the connectivity of network by connecting the remaining points within a certain distance to the skeleton network.

Second, we use the preliminary estimation of $\Delta\hat{h}$ and $\Delta\hat{v}$ on each edge to perform temporal phase unwrapping. After the unwrapping, the final estimation of the $\Delta\hat{h}$ and $\Delta\hat{v}$ are carried out via the solution of the following system:

$$\Delta\varnothing_{unw}^k = \frac{4\pi}{\lambda} \begin{bmatrix} \frac{B_\perp^1}{R^k sin\theta} & \Delta T^1 & 1 \\ & & \\ \frac{B_\perp^M}{R^M sin\theta} & \Delta T^M & 1 \end{bmatrix} \begin{bmatrix} \Delta h \\ \Delta v \\ r_{off} \end{bmatrix} \tag{4}$$

where $r_{off}$ accounts for a possible phase offset. In practice, the unwrapped phase may contain outliers due to low-quality images or unwrapping error. To mitigate the influence of possible phase outliers, we make use of iteration weighted LS-estimators to estimate the final parameters.

Finally, the velocity and the topographic error on the selected pixels can be reconstructed by using the results obtained over the edges. This is done by least squares adjustment, considering the linear velocity and the topographic error over each pixel as unknowns, and the estimated differential velocities and differential topographies as observations. As the integration only involves differential values, it is necessary to fix the velocity value of at least one-pixel reference point. The same has to be done for the topographic error. The outcomes of this stage are a velocity map and a topographic error map, whose values are relative to the chosen reference point, which can be used to compute the corresponding phase components for each of the $M$ interferograms. Both components are then subtracted from the original interferometric phase, getting a cleaned interferometric phase, which is then processed in the subsequent processing stages. This operation is particularly helpful during phase unwrapping. After this, the contribution of the phase noise, i.e., the spatial and temporal decorrelation factors, as well as the atmospheric phase, are separated based on their spatial and temporal behavior differences.

In order to combine the two datasets of RADARSAT-2 and Sentinel-1 for a more throughout interpretation, the displacement rates measured along the satellite line of sight (LOS) $V_{LOS}$ could be re-projected into a new velocity along the steepest slope direction $V_{slope}$, according to the formula proposed by Bianchini et al. [25] and Notti et al. [36]. This can be computed from the geometry of the ground surface and the satellite acquisition parameters. After projecting displacements in a common direction, it is easy to merge the ground displacements, as measured by the ascending RADARSAT-2 and descending Sentinel-1 data. In the merged displacement map, only PS/DS points of slope gradients greater than 5° were projected, and the PS/DS points with a positive displacement velocity were discarded.

*3.3. Integration*

The flowchart of the refined LSM is shown in Figure 4. The LSM obtained above was almost completely extracted from static parameters, such as curvature, altitude, slope angle, slope aspect, etc.

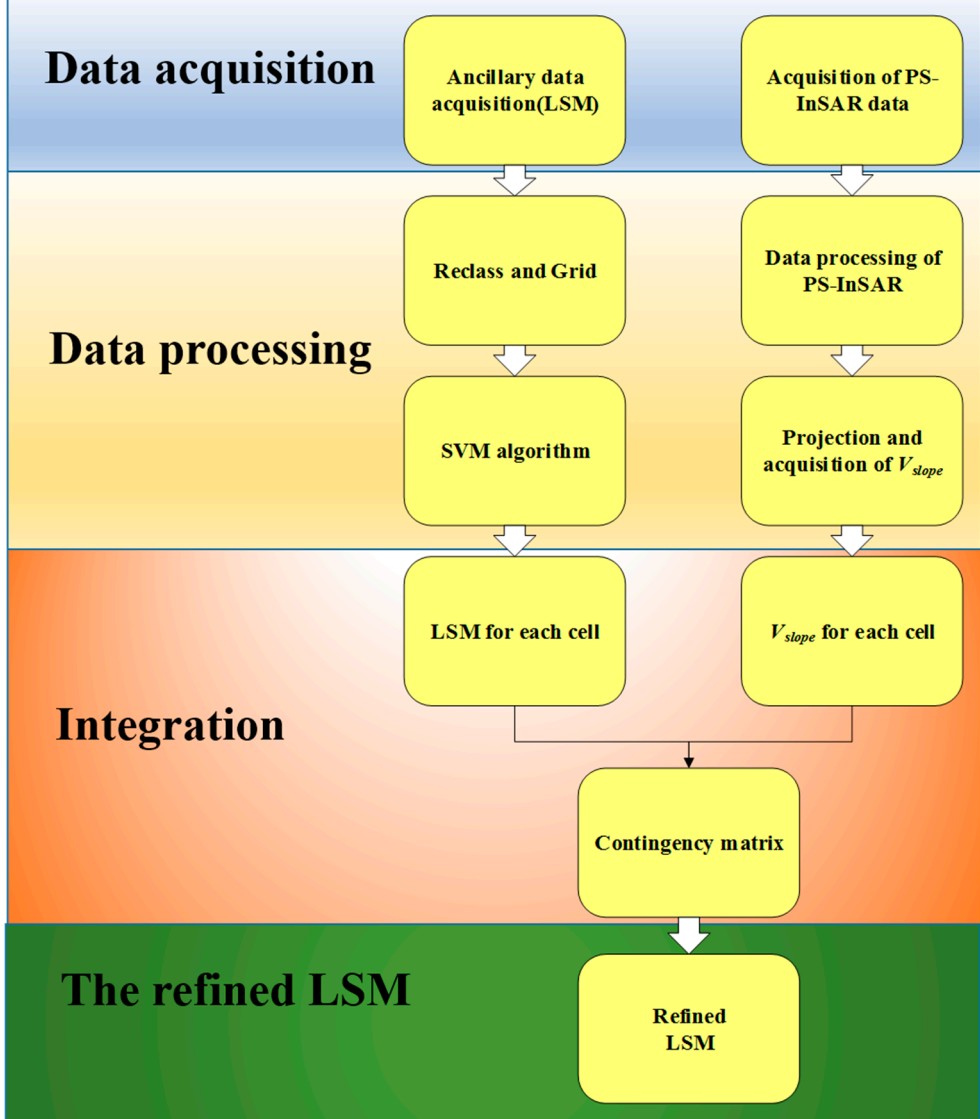

**Figure 4.** Flowchart illustrating the adopted methodology.

The PSInSAR dataset has the potential to identify slope instability signs, which may not be sufficiently clear and observable, and consequently may not be correctly identifiable through traditional geomorphologic survey techniques. Meanwhile, the PSInSAR dataset also reflects the dynamic deformation, which can easily be updated with new SAR data acquisitions. The aim of integration is to update the landslide susceptibility of slopes with active mass movements, observed by means of the PSI dataset. The direct use of PSInSAR data for the landslide susceptibility assessment can increase the susceptibility degree of those cells characterized by ground deformation.

We implemented this integration by coupling the degree of susceptibility with an average $V_{slope}$ in each cell (100 by 100 m). This work used the average velocity based on at least four PS/DS points to update the susceptibility degree of each single cell. The LSM was reclassified as low risk = 1, medium risk = 2, high risk = 3, or extremely high risk = 4, according to the natural discontinuity of probability. The $V_{slope}$ map, obtained from Equation (1), has a standard deviation of 10 mm. So,

the $V_{slope}$ map was also classified into four categories according to the standard deviation of the slope direction deformation—low deformation (0–10 mm/year), medium deformation (10–20 mm/year), high deformation (20–30 mm/year), and extremely high deformation (>30 mm/year).

It was assumed that a larger value of $V_{slope}$ represents greater landslide susceptibility in the cell (100 by 100 m). As shown in Table 2, for an area characterized by a susceptibility degree of 1, the correction value can be 0, +1, +2, or +3 with the increase of $V_{slope}$. For an area characterized by a susceptibility degree of 4, no change occurs. Based on the above method, we merged the PSInSAR and LSM with the derived correction matrix.

**Table 2.** Correction matrix applied to the landslide susceptibility maps (LSM) considering the average $V_{slope}$.

| | | $V_{slope}$ (mm/year) | | | |
|---|---|---|---|---|---|
| | | 0–10 | 10–20 | 20–30 | >30 |
| | 1 | 0 | +1 | +2 | +3 |
| LSM | 2 | 0 | 0 | +1 | +2 |
| | 3 | 0 | 0 | 0 | +1 |
| | 4 | 0 | 0 | 0 | 0 |

## 4. Results

### 4.1. Landslide Susceptibility Map

A confusion matrix is the most basic, intuitive, and simplest way to measure the accuracy of an SVM model. This work utilized a confusion matrix as a positive method to judge the performance of the classification of the two-class problem. The confusion matrix includes the information of the model classification accuracy, which identifies true positives (TN), false negatives (FN), false positives (FP), and true negatives (TN). The confusion matrix illustrated that the accuracy of the model (TP + TN/TP + TN + FN + FP) was 77.29%.

The receiver operating characteristic (ROC) has been widely used in many fields to evaluate algorithm performance in LSMs. The ROC provides a graphical plot in which the binary classifier method performance is determined. To validate the dataset, the true positive rate (TPR) and false positive rate (FPR) were calculated by using a confusion matrix. The vertical axis of the ROC curve is the TPR, and the horizontal axis is the FPR. Using the validation dataset, the results of the area under the curve (AUC) are shown in Figure 5. This figure illustrates that the RBF function has high accuracy in the study area since the area of the AUC using the validation dataset is 0.89.

After the SVM model was successfully trained with the training set, it was used to calculate the landslide susceptibility indices for all pixels in the study area. Using the map algebra in ArcGIS, the LSM of the study area was calculated and reclassed into four susceptibility levels. The total number of pixels gave 2,394,053 cells. The overall LSM of Bijie City is shown in Figure 6. It was found that 25.26% of the total area had low susceptibility, 30.21% had medium susceptibility, 41.30% had high susceptibility, and 4.23% had very high susceptibility. The assessment of the contribution of the conditioning factors to the models is also an important step in landslide analysis. In general, topography, geology, hydrology, and geomorphology are widely accepted as the conditioning factors in most landslide susceptibility models. Karst terrains are widely distributed in Bijie and lead to a great number of unstable slopes. Meanwhile, intense artificial activities, such as coal mining, have further disturbed the stability of the slopes. As shown in Figure 7, the most landslide-conditioning variables (LCV) are mining disturbance, elevation, and slope angle. This showed that the landslide susceptibility of Bijie was mostly triggered by disturbances caused by mining coal. In fact, coal-mining operations lead to incremental vertical ground movement. Furthermore, in the rainy season, rain can cause water flux and soil saturation, contributing to the increase of landslide susceptibility. The importance of the slope angle is expected, as it usually represents one of the most important predisposing factors for landslides.

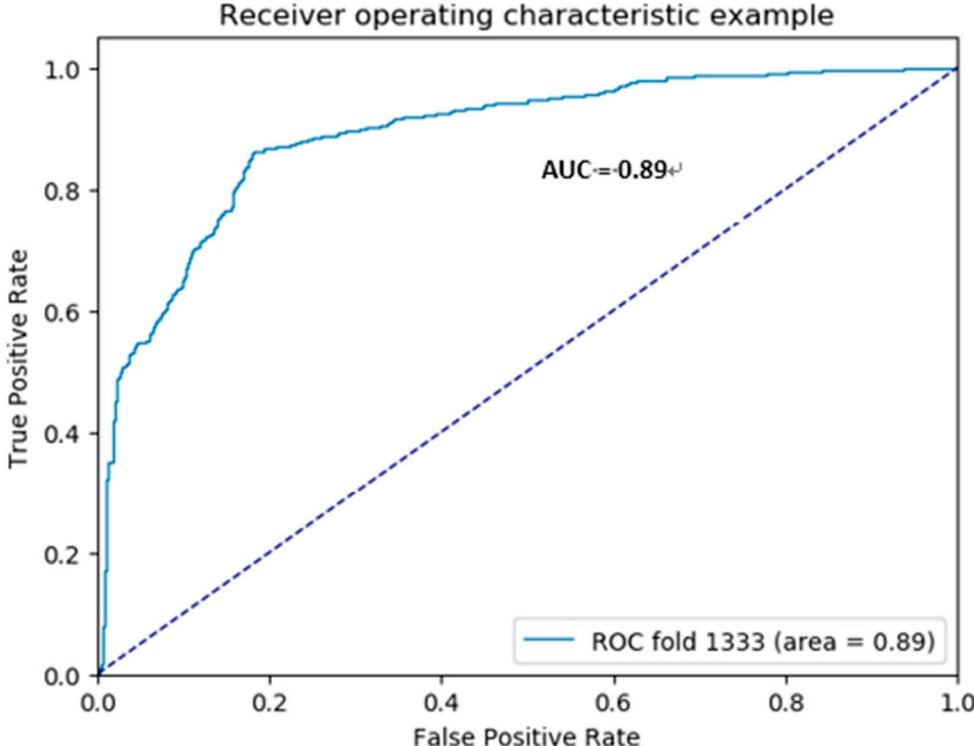

**Figure 5.** Receiver operating characteristic (ROC) curve for evaluating the performance of the adopted model.

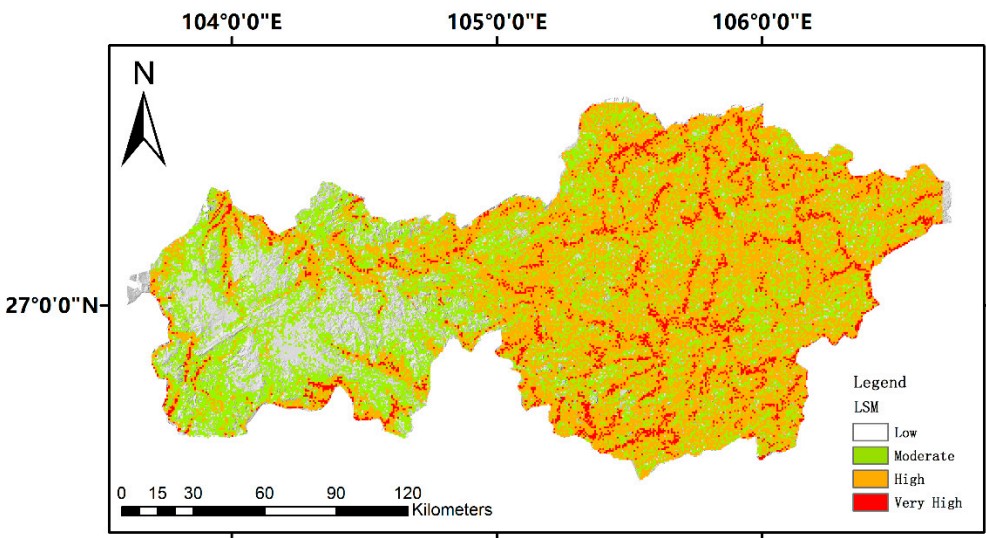

**Figure 6.** The LSM obtained by applying the support vector machine (SVM) algorithm.

As shown in Figure 8, Nayong country and Dafang country are used as examples to illustrate the relationship between LSM and influence factors. As mentioned earlier, mining is the most important factor in the SVM analysis. The violet region is the distribution of coal seam, which basically spreads in the high and very high. The high and very high landslide susceptibility region distributed around the coal regions. The river and rainfall are also important factors that are discussed later in the paper. After the rainstorm or long-term rain, the continuous drought will cause dangerous landslides. On the one hand, there are unloading cracks at the rear edge of the mountain, and the precipitation infiltrates into the unloading cracks to cause the increase of water pressure. On the other hand, the mountain itself is in a hard and soft structure, and the precipitation infiltrates into the joint fissures, dissolution pipes, and

other channels, resulting in the low strength of the lower soft rock. Under the dual action of unloading fissure water filling and base softening, the rock mass stability is reduced and landslide occurs.

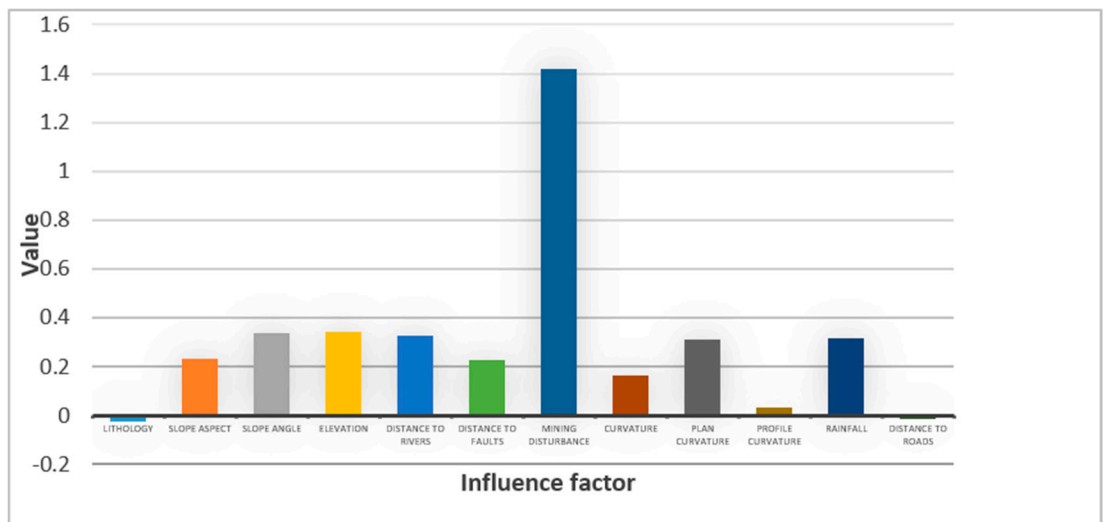

**Figure 7.** Example plot illustrating the variation of the relative importance of parameters affecting landslide susceptibility.

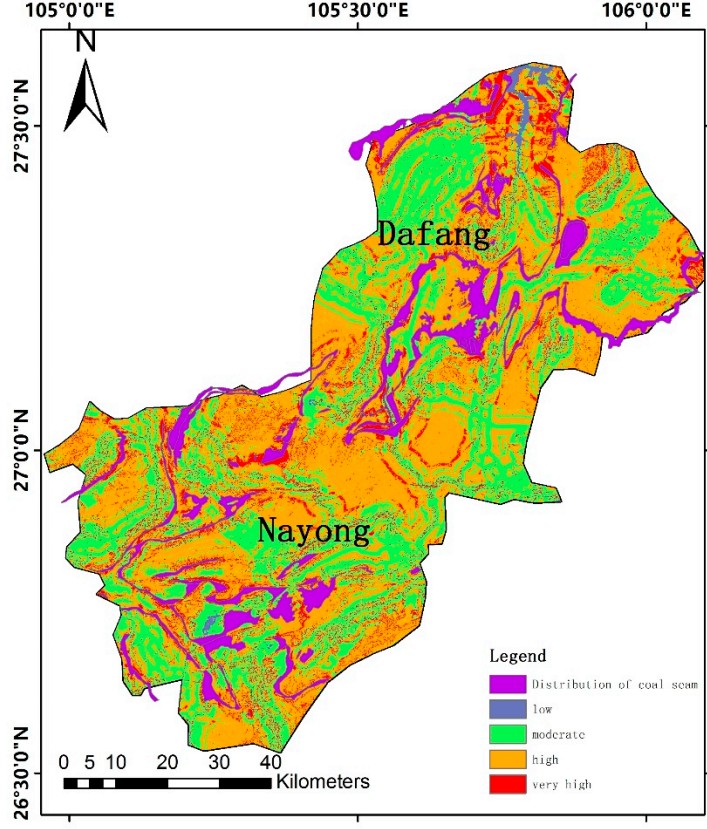

**Figure 8.** Comparison of distribution of coal seam and LSM.

*4.2. PSInSAR*

This work performed a PSInSAR analysis for the whole region of Bijie with RADSAT-2 and Sentinel-1 datasets. In total, 28,666,959 PS/DS were available from the RADARSAT-2 dataset, and the density of PS/DS was 1067.5 points/km$^2$. Furthermore, 4,777,826 PS/DS points were available from the Sentinel-1 dataset, and density of PS/DS was 177.9 points/km$^2$.

We evaluated the geometrical visibility of the area of interest based on the R-indices, using the approach proposed by Notti, with respect to the morphology and the acquisition geometry of the available satellite system.

R-index (RI) maps of the ascending RADARSAT-2 and descending Sentinel-1 data were generated to assess the probability of movement along a slope detectable from two opposite directions, and a combined geometrical visibility map was formed by merging the ascending and descending R-index (RI) maps together. As shown in Figures 9 and 10, RI values below 0.4 (class 1, 2, and 3) indicate that the slope geometry is not favorable for the satellite acquisition geometry; this accounted for 22% of the study area. RI values above 0.4 but below 0.6 (class 4) represent slopes with an acceptable geometry with respect to the satellite line of sight (LOS) or flat area, accounting for 40% of the study area. RI values above 0.6 (class 5) indicate that the slope direction of the area is approximately parallel to the satellite LOS direction; this accounted for 38% of the study area, and was the most suitable for landslide investigation, using the PSI technique. Figures 9 and 10 illustrates that most of Bijie has good geometrical visibility after merging the ascending RADARSAT-2 and descending Sentinel-1 data.

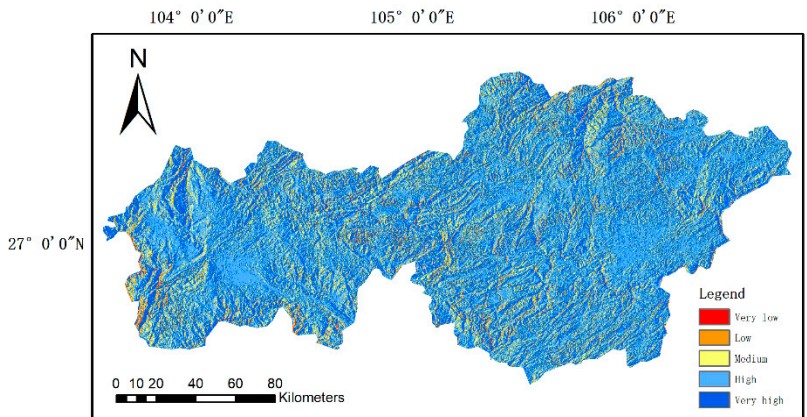

**Figure 9.** Combined geometrical visibility map (R-index map) with ascending RADARSAT-2 and descending Sentinel-1 datasets.

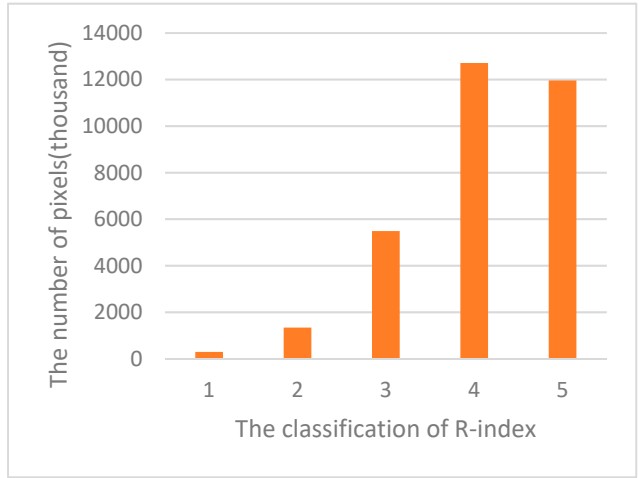

**Figure 10.** The histogram of R-index.

Figure 11a shows the ground deformation velocity ($V_{LOS}$) map along the satellite LOS, illustrating that there were obvious ground deformation phenomena (landslides and subsidence). Especially in the southeast region of the study area, which is the major coal mine area of Bijie, the maximum absolute displacement velocity $V_{LOS}$ reached up to 73.7 mm/year. Figure 11b depicts the ground deformation velocity ($V_{slope}$) map along the slope direction. Because PS/DS points with a positive

displacement velocity were discarded, along with PS/DS points in flat areas, the $V_{slope}$ map has a lower PS/DS density than the $V_{LOS}$ map. Considering the standard deviation of the velocity of the PSI dataset, this work chose the stability threshold of 0–10.00 mm/year. Figure 11b highlights that 66% of the PS/DS population of a $V_{slope}$ greater than 27 mm/year is located in Eastern Bijie. The distribution of PS/DS points of great displacement is identical to the distribution of coal mine reserves.

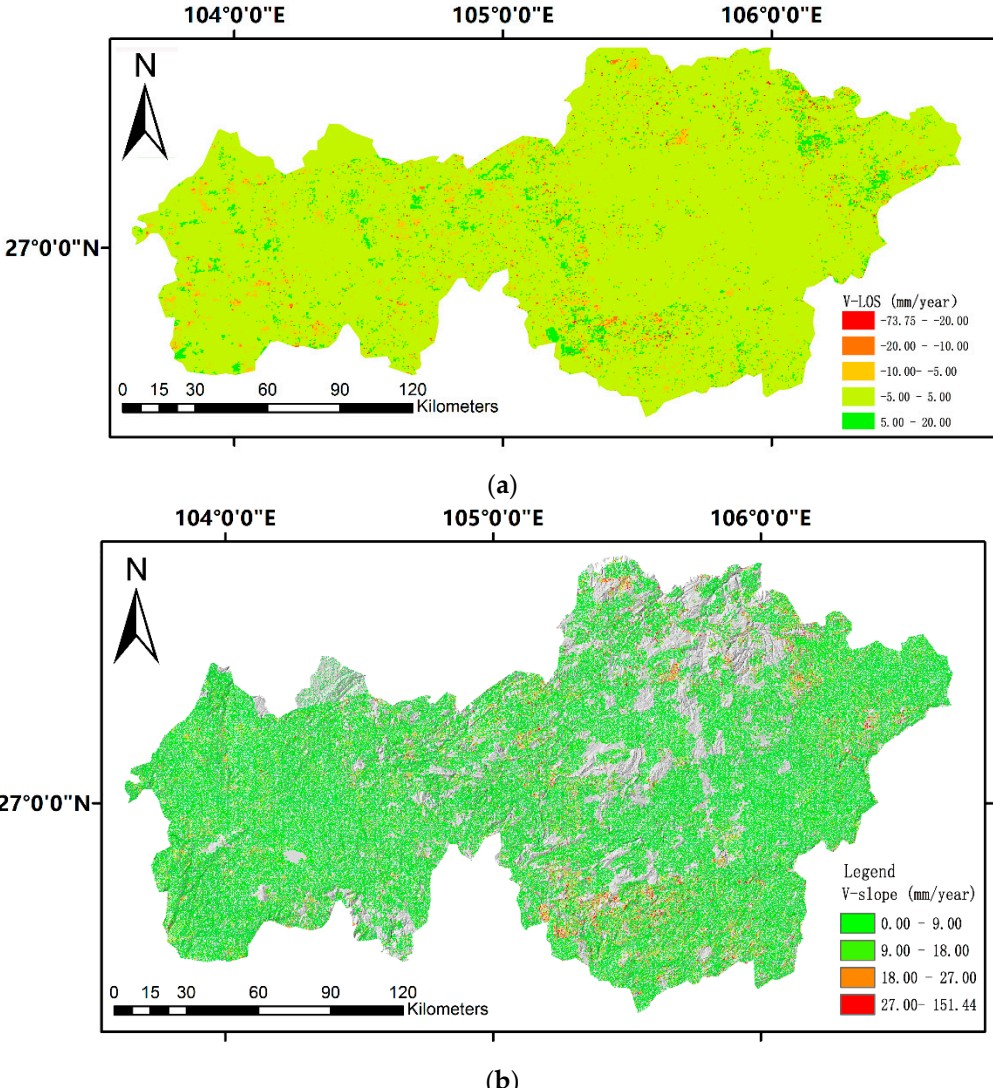

**Figure 11.** Ground deformation velocity maps obtained using the merged $V_{LOS}$ dataset (**a**) and $V_{slope}$ dataset (**b**).

### 4.3. Integration

The first step of the integration procedure was to resample $V_{slope}$ on each PS/DS point to an average $V_{slope}$ in each cell (100 by 100 m). The average $V_{slope}$ was no longer related to a single PS/DS point, but rather to a 100 by 100 m cells. In this work, the total number of these cells was 2,394,053, which covered the whole area of Bijie. After generating a ground deformation velocity map for each cell, a correction matrix was utilized to refine the LSM.

The refined LSM was characterized by the reduction of the area with the lowest susceptibility level and an increase in the area with the highest susceptibility level. Specifically, four levels in the new LSM (from low to high) exhibited noticeable changes. For example, in the refined LSM, the percentage of class 1 (low to null susceptibility) was 14.33%, the percentage of class 2 (moderate susceptibility) was 33.25%, the percentage of class 3 (high susceptibility) was 45.87%, and the percentage of class 4

was 6.55% (Figures 12 and 13). In total, the results show that the susceptibility degree increased in 56.41 km$^2$ of the study area, and 80% of the increased susceptibility degree were caused by coal mining. Coal mining results in surface deformation and increases landslide susceptibility.

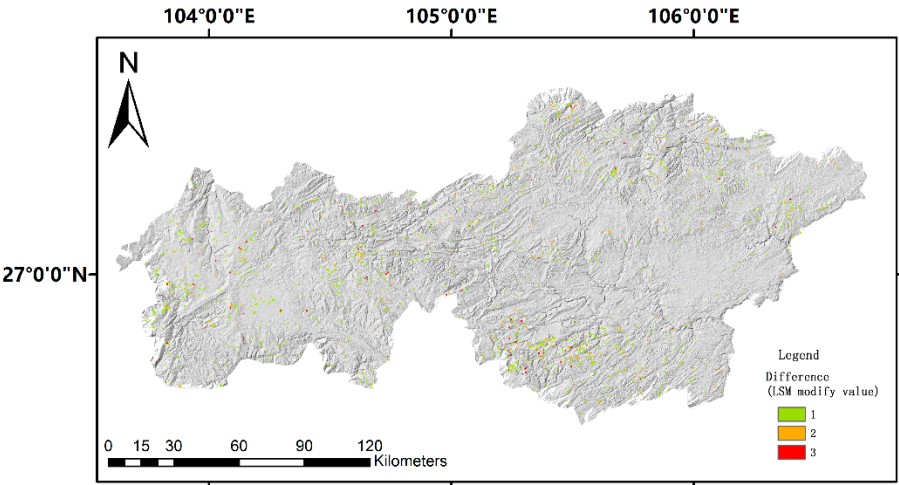

**Figure 12.** The difference between the original LSM and the new LSM after the application of the correction matrix.

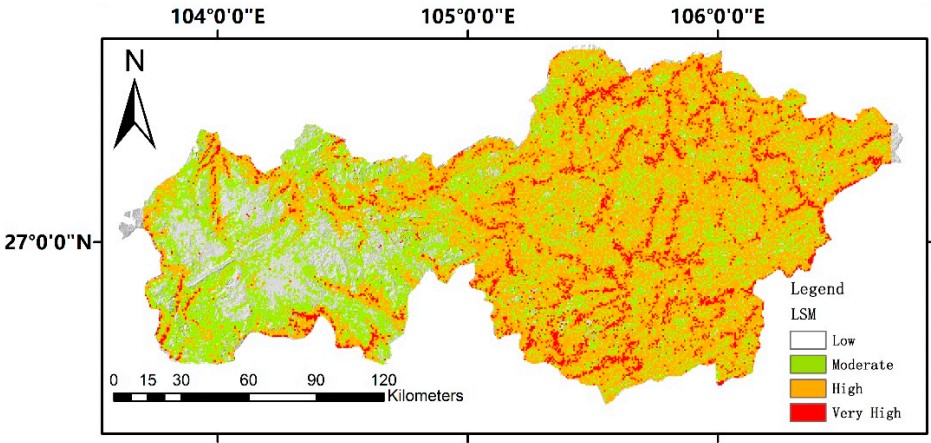

**Figure 13.** The new LSM after the application of the correction matrix.

For example, Nayong country and Dafang country are chosen to explain changes after modification. As seen in Figure 14, there are 2,541,519 pixels in the Nayong country and Dafang country, which are divided to 30 m by 30 m cells. As shown in Figure 14, the modified pixels are 3.91% of the total pixels. Most of the pixels are distributed in the southeast of Nayong and the northwest of Dafang and on both sides of the rivers. According to Figure 7, mining distribution is the most important parameter affecting landslide susceptibility, which means mining distribution has a great influence on the initial LSM. The key factor is that mining activities destroyed the underground structure and changed the stress conditions. When affected by river erosion or rainfall, this effect will be more obvious. That's why we modified the pixels to present a distribution along the river.

Next, we briefly discuss the details of two specific cases of the Zongling and Gaodian Regions. In 2019, we conducted a field geological survey of these two regions.

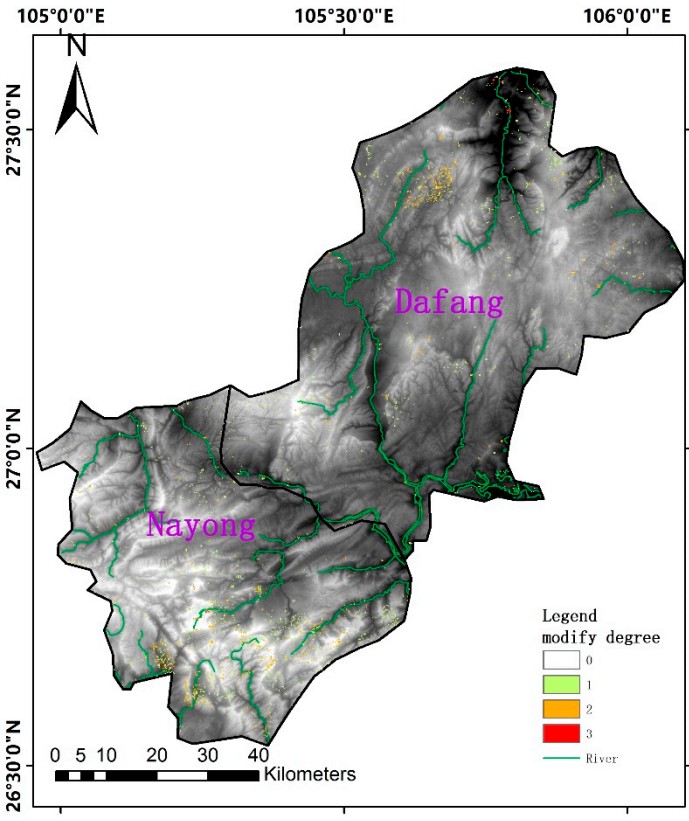

**Figure 14.** The modify degree in Nayong and Dafang.

### 4.3.1. Results of Zongling

The first region, Zongling, is located in eastern Bijie. Zongling is a small town of nearly 10,000 people, and it is strongly influenced by landslides. As the main industry in the town, the coal mining operation is extremely intensive—there are four coal mining enterprises, with 21 mining sites in the town. In the north of Zongling, there is a mountain called Mazongling mountain. Intensive coal mining operation has triggered at least eight slow-moving landslides on the south slope of Mazongling mountain. The largest landslide has an area of 189,680 m$^2$. Furthermore, nearly 5000 people, including students in Zongling Middle School, live under the threat of this group of landslides. Therefore, the local government started to relocate these residents starting last year.

Most of the LSM cells located in the area affected by the largest landslide showed a high or very high susceptibility, as predicted using the SVM algorithm. The LSM showed a low susceptibility in flat and residential areas. The PSInSAR velocity field confirmed that the largest landslide is still active, with $V_{slope}$ values higher than 40 mm/year. In this case, the correction matrix could be beneficially applied to update the LSM. The new LSM showed these changes, adding the area affected by ground deformation in the PSI result. After this step, updating the landslide inventory map and extending the boundary of the biggest landslide is recommended. On the field survey, some phenomena related to landslide were observed—for example, wall cracking and rock cracking—which shows that the landslide area is active. The new LSM (Figure 15c) assigned a higher susceptibility degree to some new landslides, considering those areas affected by ground deformation highlighted in the PSI map.

Rainfall and mining activities are the most important factors when considering the cause of landslides in the region. In fact, the coal mine is the economic source of the local government. Coal is used for fuel by local residents. Coal-mining activities form goafs under the mountains. Ground subsidence and ground cracks occur, and subsidence deformation will cause mountain stress. The frame borders in Figure 15 are the distribution of coal seams. The main regions of landslides occur in the coal seam.

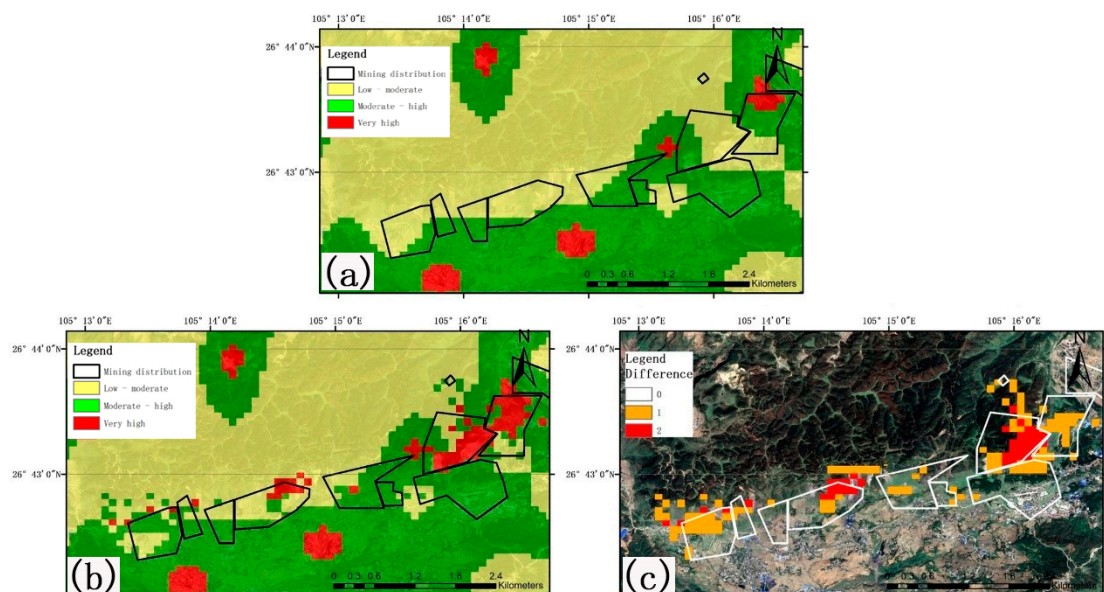

**Figure 15.** (**a**) Original LSM of Zongling; (**b**) $V_{slope}$ ground deformation velocity map; (**c**) new LSM obtained after the application of the correction matrix.

As shown in Figure 16, the PSI results and landslide ranges in Zongling are good examples to explain the necessity of refined LSM. We choose three points to illustrate how rainfall and intense mining activities affect landslides in this area. During the rainy season (May to August), there is a positive correlation between deformation and rainfall, but in the dry season (November to March), there is no obvious relationship between deformation and rainfall (Figure 17). In November 2017 to March 2018, the rainfall is very low, but the velocity of deformation is up to –0.6 mm/day, which proves that there are factors influencing deformation beyond rainfall. According to these factors in the SVM model, some factors are static and relatively stable, such as lithology and slope. This anomaly was proved to be caused by mining activities. Coal mining has a dynamic and continuous effect on the deformation. However, rainfall is seasonally changing and causes seasonal deformation. Similarly, points B and C show the same trend as point A (Figure 18).

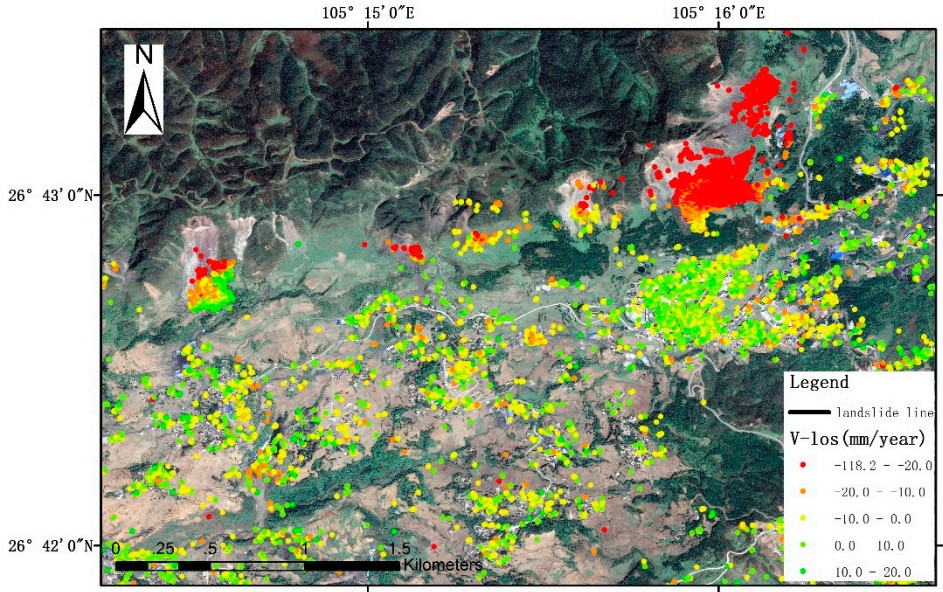

**Figure 16.** The Persistent Scatterer Interferometry (PSI) result and landslide range in Zongling.

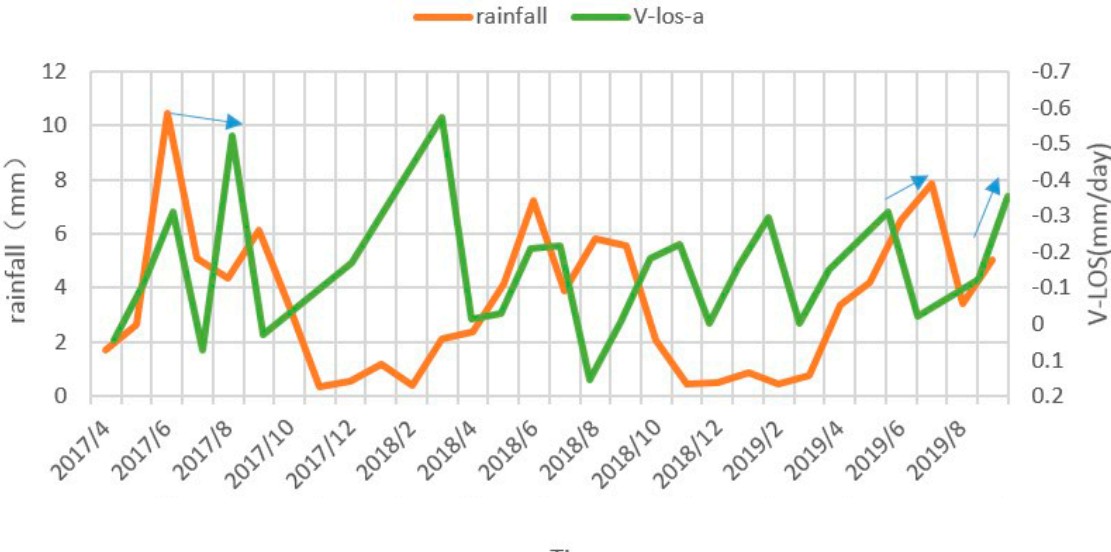

**Figure 17.** The comparation of rainfall and $V_{LOS}$ of point A.

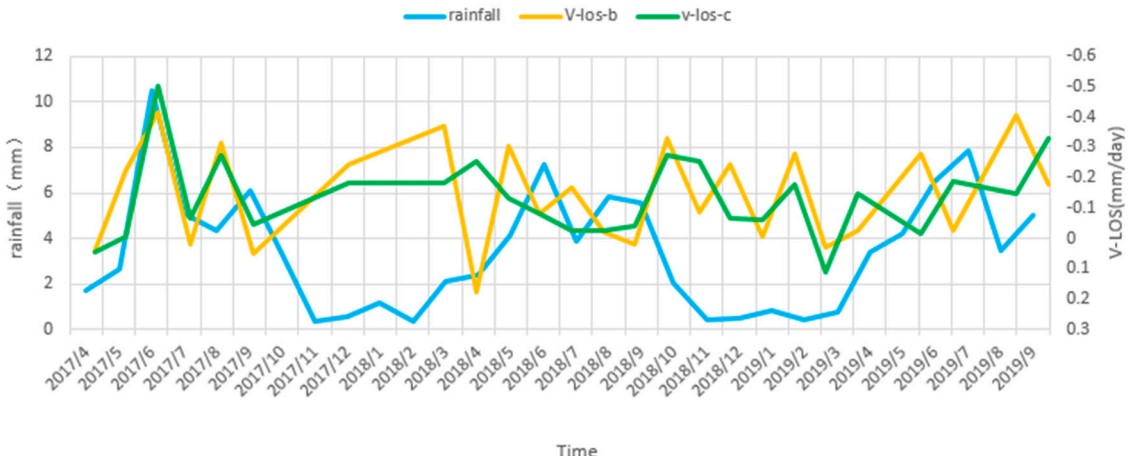

**Figure 18.** The comparation of rainfall and $V_{LOS}$ of points B and C.

### 4.3.2. Results of Gaodian

The second region is located in Dafang County, in Eastern Bijie, and is called Gaodian. This town is close to the Luojiao river. The largest landslides are located in the southwest of the town, in Shiwei Village, and in a coal mining area situated north of the town. Affected by the Luojiao river, the landslides may dam the lake. This area was chosen to test the proposed approach because, in the available landslide inventory map (LIM), few landslides are mapped. On the contrary, PSInSAR data showed the main ground deformation on the largest landslide, which was also verified by our field investigation. In fact, the SVM algorithm reduced the susceptibility degree of this area due to errors including FN and FP (Figure 19a), which is characterized by high ground deformation velocities. The map of the difference between the classic LSM and the updated LSM (Figure 19c) shows that several landslides were not mapped in the LIM used to create the LSM (Figure 19d). Considering some previously undetected phenomena in the LIM, this method can be used to update LSM in the study area at the same time.

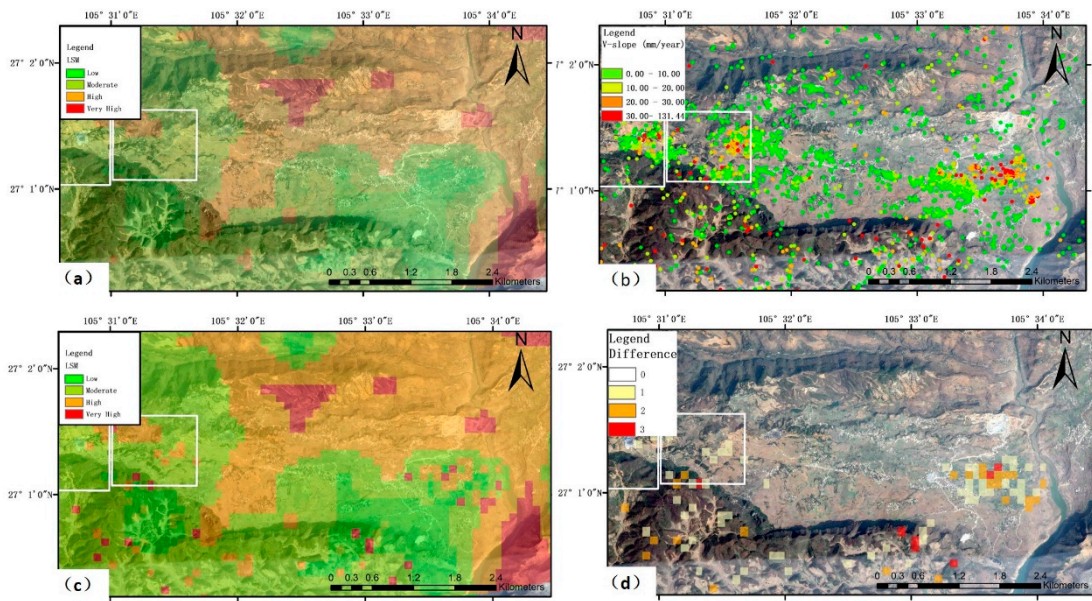

**Figure 19.** (**a**) Original LSM of Gaodian; (**b**) VSlope ground deformation velocity map; (**c**) new LSM obtained after the application of the correction matrix; and (**d**) difference between the original LSM and the new LSM.

In the landslide of Gaodian, point D is chosen to illustrate the influence of factors on the LSM. (Figures 20 and 21) According to the example in Zongline, rainfall has a positive impact on landslides and shows a delay to the deformation. It is found the same situation at the point D. However, point D is far away from the mining area, so it must be the other factor that causes abnormal deformation during the dry season. Because point D is close to the Luojiao river, we speculate that the deformation may be related to groundwater level and river erosion. So, we plan to collect the data of groundwater level and then use the other data to analyze the causes of this situation, in a follow-up work.

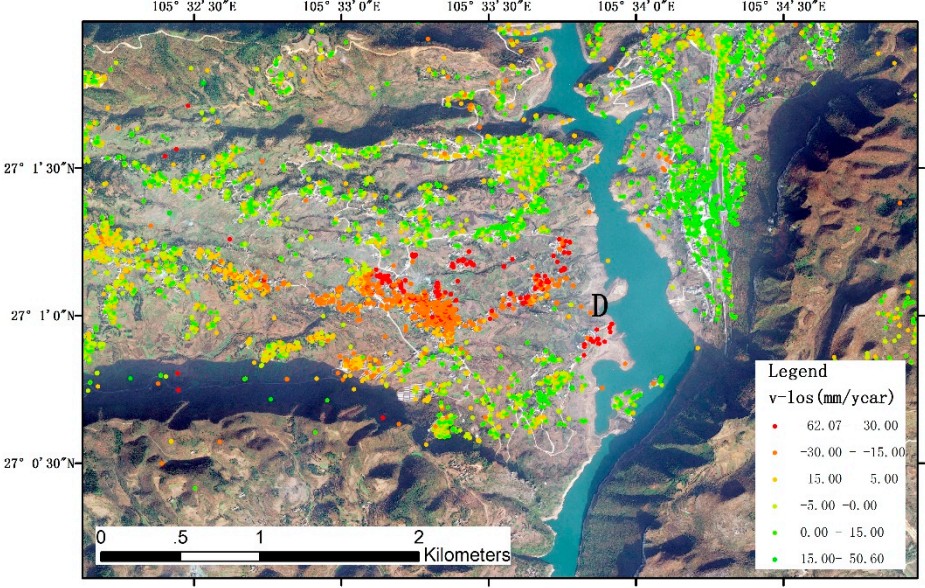

**Figure 20.** The PSI result and landslide in Gaodian.

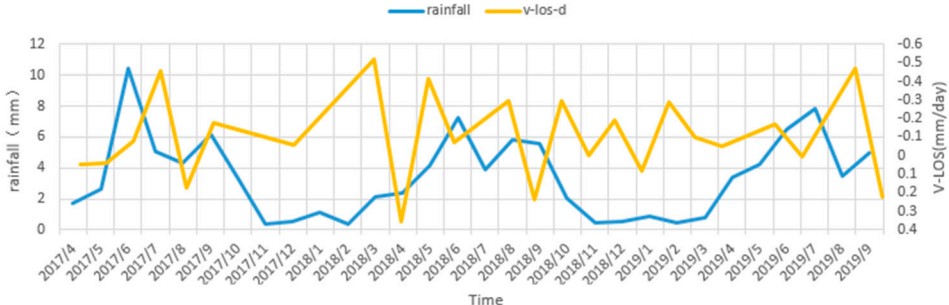

**Figure 21.** The comparison of rainfall and V-los of point D.

## 5. Discussion

### 5.1. Potentials and Limitations of PSInSAR for Karst Region

The results reveal that the PSInSAR map can improve the susceptibility ranking of a portion of the Karst Region and produce more-reliable landslide susceptibility maps as compared to the stickily geomorphological approach. It should be verified that these landslides, which can be monitored by using the PSI datasets, are extremely slow-moving and very slow-moving landslides with mean displacement velocities (LOS) below 1.0 m/year. PSI can hardly measure landslides with a comparably large rate of movements or strong seasonal variations due to the temporal decoherence. This excludes from the analysis all the potential PS candidates whose mean deformation velocities suffer a displacement acceleration higher than the technique limitation. Pixel offset tracking, based on amplitude cross-correlation between SAR images, has been proven to be an effective tool for robustly measuring large displacements with velocities (LOS) of about 10 m/month and supplying measurements of two-dimensional deformations. The offset tracking method should be included to update the information on this kind of landslide in the available landslide inventory map (LIM).

Topography is limiting the application of this technology for the Karst Region, in particular due to geometrical limitation related to overlay and shadow. Another aspect related to the topography and the special SAR viewing is that PSInSAR is only sensitive to movements taking place along the line of sight (LOS) direction. If the LOS orientation is not favorable in relation to terrain deformation, the number of detected points might be reduced. This problem can be mitigated by combining ascending and descending orbits, e.g., RADARSAT-2, with ascending orbits and SENTINEL-1 with descending orbits in this study. Based on our analysis it is evident that best results are achieved by the complementarity use of different acquisition geometries and different sensors.

Vegetation in the Karst Region is another limiting factor that should be overcome, this work implemented a distributed scatterer technique, which fully utilizes InSAR stack data by forming an optimum interferogram network and implementing spatially adaptive filtering, to reduce noise and improve the density of distributed scatterers in sparsely vegetated landscapes. Although PS/DS points from the C-band dataset exhibited a lower density in vegetated areas with respect to those from the L-band dataset, the RADARSAT-2 dataset in the Extra Fine mode has a higher temporal sampling of acquisitions with a swath width of 120 km, which reduces the effect of temporal decorrelation, thus allowing the detection of displacement over landslides covered by rangeland, pasture, shrubs, and bare soils.

### 5.2. Refinement of LSM by Using PSInSAR

PSI identifies only features with observed surface movements and misses landslides temporally without movement activity or not appropriate to be analyzed with this technology, e.g., because of rapid and sporadic rate of motion. On the other hand, the LSM only uses some definably deform features, which has limitations in estimating recent movement activity of individual landslides. By combining an LSM with PSI, the degree of landslide susceptibility in those areas affected by ground-land deformation

can be updated and corrected, and thus increasing the reliably of the LSM. So, we analyzed two selected parts of the study areas, including Zongling Town and Gaodian Town.

To update the LSM by using PSI data, we planned to acquired RADARSAT-2 images over the previous two years, so as to continuous updated the PSI dataset in the next year. In addition, the descending Sentinel-1 dataset will be continuously updated. We also started to collect L-band ALOS-2 SAR data, which can keep a relatively high coherence level over densely vegetated areas in the Karst Region, and subsequently generate differential interferograms for the purpose of investigating landslides with mean displacement velocities (LOS) above 1.0 m/year. We are also currently exploring methods to merge the C-band PSI dataset and L-band D-InSAR dataset to produce a combined V_slope map.

In Bijie, the local government invests a great deal of financial and human resources to update the LIM based on remote sensing and field investigation. Optical images (aerial photographs and optical satellite images) are also utilized to identify the terrain features responsible for landslides, as well as LiDAR-derived topographic information, to characterize and differentiate landslide morphology and activity. Meanwhile, automatic monitoring equipment, such as digital rain gauge recorders, shape acceleration arrays (SAAs) for deep-displacement monitoring, and low-cost GNSS (Global Navigation Satellite Systems) equipment, will be gradually installed on the major landslides in the next few years. After the LIM is updated with detailed information obtained from remote sensing and field investigation, it can be used to produce a more reliable LSM. Precipitation data from automatic meteorological stations, other than the monthly rainfall distribution used in this work, will also greatly improve the effectiveness of the LSM. This work produced a preliminary version of the LSM, which will become more and more reliable in evaluating the risks for people and infrastructure threatened by landslides, thanks to the updates provided by new PSI datasets and field investigations.

## 6. Conclusions

This work focused on the dynamic updating of a susceptibility map, using PSI data in areas of intense mining activities. Coal mining operations in the Karst Region have increased landslide susceptibility greatly in a short time and triggered a large number of geological disasters. The PSI technique was used to update the available landslide inventory map in order to generate a more reliable LSM. It is often time-consuming to generate a more reliable LSM after updating the available landslide inventory map by using PSI datasets in broad areas. To update the LSM in time, this paper presented a new approach to update landslide susceptibility maps by using PSI data directly.

A PSInSAR analysis was implemented in the whole region of Bijie with ascending RADARSAT-2 and descending Sentinel-1 datasets acquired between 2017 and 2019. A DSI technique, which can considerably increase the point target density in sparsely vegetated landscapes, was utilized to produce the landslide displacement map.

The preliminary LSM of Bijie was produced with a 100 by 100 m cell resolution, using an SVM algorithm in the sklearn package of Python. The landslide displacement map, produced by the PSInSAR analysis, was projected to the direction of the steepest slope and resampled to the same cell in the LSM, to update the original LSM. This paper implemented a contingency matrix to adjust the susceptibility degrees of cells, with the assistance of the PSI data.

A fairly good AUC value (0.89) of the ROC curve revealed that the PSInSAR map could improve the accuracy of the LSM. Finally, a comparison between the original and refined LSM in two specific areas revealed that the proposed method is capable of producing more-reliable landslide susceptibility maps in time.

**Author Contributions:** C.S., C.X., Z.F., and H.F. performed the experiments and produced the results. C.S., C.X., and H.F. drafted the manuscript. B.Z., W.O., H.B., P.C. and A.M. contributed to the discussion of the results. Y.Z., K.W., and H.L. helped to collect and analyze the SAR data. All authors conceived of the study and reviewed and approved of the manuscript.

**Funding:** This research was funded by the National Key R&D Program of China (Grant No. 2018YFC0809400), the Fundamental Research Funds for the Central Universities (Grant No. 2015ZCQ-LX-01), the Science and Technology project of State Grid (Grant No. GCB17201700142), and the National Natural Science Foundation of China (Grant No. 41571328, U1710123).

**Acknowledgments:** RADARSAT-2 data were provided by MDA, Canada. Sentinel-1 data were provided by ESA, Eu. The one arc-second SRTM DEM was freely downloaded from the website http://e4ftl01.cr.usgs.gov/MODV6_Dal_D/SRTM/SRTMGL1.003/2000.02.11/. Three-arc-second SRTM DEM was freely downloaded from http://www2.jpl.nasa.gov/srtm/cbanddataproducts.html.

**Conflicts of Interest:** The authors declare no conflicts of interest.

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
