# Peer review of "Refinement of Landslide Susceptibility Map Using Persistent Scatterer Interferometry in Areas of Intense Mining Activities in the Karst Region of Southwest China"

_remotesensing, doi:10.3390/rs11232821_

Round 1
Reviewer 1 Report
The authors complied with all my comments.
In my opinion the paper is ready for publication.
Reviewer 2 Report
The Authors improved the comments, therefore as it is the writing can be accepted.
This manuscript is a resubmission of an earlier submission. The following is a list of the peer review reports and author responses from that submission.
Round 1
Reviewer 1 Report
The paper “Update of landslide susceptibility map using Persistent Scatterer Interferometry in areas of intense mining activities in the Karst Region of Southwest China” deals with the description of a method to produce a landslide susceptibility map using SVM and interferometric satellite monitoring.
The English language is not always correct or understandable. The methodology is not properly explained (especially how the starting map is obtained starting from geomorphological and geological data). However the major limitation is that in my opinion the methodology and results of the paper are not sufficiently novel or interesting to be published in Remote Sensing.
A list of more detailed revisions follows:
Lines 56-58: This sentence is not clear and should be rephrased: “The sloping ground surfaces impose additional subsidence effects, and the tendency of increased deformations angling towards valleys and slopes is a common problem that arises when mining in sloping grounds”. In particular you should explain why you say so.
87-88: Please rephrase this sentence and check English.
89-91: the main verb is missing in this sentence.
93: reservation?
111: maybe fig 1c?
126: spell RADARSAT-2
143: when you cite Notti you should include the date and a proper reference
151: THIS accounted for 70%
152: this accounted FOR
173-182: this part is repeated. Delete it.
185 and 186: delete “of”
190: please explain what is an L1-norm based method
197: I suggest using PS/DS instead of just DS (like you did in other parts of the text), or to write a more generic “measurement points” which you can abbreviate into MP.
199: this formula is a very basilar one and has not been proposed by Bianchini or Notti, it is just a simple trigonometric calculation. Please remove these two references.
236: this figure is not referenced to in the text and should probably be placed later.
239-240: how exactly? It is important to understand how you retrieved this map. At line 281 you mention 12 factors but give no explanation.
251-252: what do you mean with natural discontinuity of probability?
319-320: this sentence about the karst nature of the area is not linked to the rest. What is your point here?
328: the colours of fig 10 are very similar to each other. There is also a 0% that should be removed and explained. Apparently there is a -1% as well.
390: replace monitoring with monitored
Reviewer 2 Report
Dear Editor,
As I understand, this manuscript for "Update of landslide susceptibility map using Persistent Scatterer Interferometry in areas of intense mining activities in the Karst Region of Southwest China "represents landslide susceptibility map using SAR and GIS techniques in the Karst area of Southwest China. I think the theme of this research is not coinciding with subject of Remote Sensing. The main results obtained this paper are not of interest to specialists studying the recent surface deformations using InSAR time-series analysis and GIS method. Indeed I am not recommend this manuscript should be published this journal for improving research topics for normal method such as SVM and not enough to analysis of InSAR time series in this area.
First of all, update of landslide susceptibility map using InSAR time-series analysis is not affected by result of this kind of research because of same figures, when I compare to before and after update of this result such as figure 6 and figure 8. Study area is too big for analysis landslide susceptibility, even the author did not process specific small area subsidence result. I am sure why the author use the InSAR time series technique for applying this research, this technique did not effect to any result of analysis of this study. The author should focus to specific landslide area or large region for making landslide susceptibility map, readers very confused this work.Reviewer 3 Report
In this work, the Authors show a new approach to to generate more reliable landslide susceptibility maps (LSM) using Persistent Scatterer Interferometry (PSI) data. In order to do this they provide a contingency matrix to adjust the susceptibility degrees of cells with the assistance of the PSI data. Finally, they state that the comparison between original and refined LSM in two specific areas revealed that the proposed method can produce more reliable landslide susceptibility maps.
As is the writing is unclear in many places. I suggest rewriting the Methodology section in which they put some results (interferometric results and LSMs). Furthermore, in the Results section they put some methodology. Moreover, even the Results for the two sites (Zongling and Gaodian) are not clear and too much briefly described, they don’t explain the problem that affects the sites. Finally, the Discussion section has to be rewrite completely, because it is a simple listed of possible techniques.
That said, with some work, I feel that this paper would be rejected.
Other comments:
Acronyms’ meaning should be explained when they are used for the first time
Many Figures aren’t mentioned in the text
Many Figures are wrong mentioned in the text (i.e. Figure 3, Figure 5,…..)
There are a lot of repetition in the text (i.e. lines 173-182)
Some references are wrong (i.e. [37])
Figure 5: please use a more detailed interval values (velocity: i.e. 0-5; 6-10…..)
Line 222: the state that the Authors use 12 predisposing factors, but they listed only 11.
Line 226: Please insert the metereological stations
Figure 6: Which is the criterion used to classify the susceptibility (natural break, quartile, equal interval…)?
Figure 10: How the Authors did obtain the results showed in the figure?
English language revision is advised. It would likely help if a native English speaker could read-over once before submitting it again.